

# Caprellid amphipods (*Caprella* spp.) are vulnerable to both physiological and habitat-mediated effects of ocean acidification

Emily G. Lim[1] and Christopher D.G. Harley[1,2]

[1] Department of Zoology, University of British Columbia, Vancouver, British Columbia, Canada
[2] Institute for the Oceans and Fisheries, University of British Columbia, Vancouver, British Columbia, Canada

## ABSTRACT

Ocean acidification (OA) is one of the most significant threats to marine life, and is predicted to drive important changes in marine communities. Although OA impacts will be the sum of direct effects mediated by alterations of physiological rates and indirect effects mediated by shifts in species interactions and biogenic habitat provision, direct and indirect effects are rarely considered together for any given species. Here, we assess the potential direct and indirect effects of OA on a ubiquitous group of crustaceans: caprellid amphipods (*Caprella laeviuscula* and *Caprella mutica*). Direct physiological effects were assessed by measuring caprellid heart rate in response to acidification in the laboratory. Indirect effects were explored by quantifying caprellid habitat dependence on the hydroid *Obelia dichotoma*, which has been shown to be less abundant under experimental acidification. We found that OA resulted in elevated caprellid heart rates, suggestive of increased metabolic demand. We also found a strong, positive association between caprellid population size and the availability of OA-vulnerable *O. dichotoma*, suggesting that future losses of biogenic habitat may be an important indirect effect of OA on caprellids. For species such as caprellid amphipods, which have strong associations with biogenic habitat, a consideration of only direct or indirect effects could potentially misestimate the full impact of ocean acidification.

## INTRODUCTION

Human activities are releasing substantial and increasing quantities of carbon dioxide into the atmosphere, and a sizeable fraction of these emissions are absorbed by the oceans (*Sabine et al., 2004*; *Sabine & Tanhua, 2009*). As anthropogenic $CO_2$ dissolves in seawater, it alters carbonate chemistry and causes a decrease in pH (*Feely et al., 2004*; *Quéré et al., 2009*). This change in oceanic chemistry—termed ocean acidification (OA)—has resulted in a decrease of pH by 0.1 units since pre-industrial times, with further decreases of 0.3–0.5 units predicted within this century (*Stocker et al., 2013*). Geological records suggest that the current rate of OA greatly exceeds anything observed the fossil record over the last 300 million years (*Hönisch et al., 2012*). Because OA is projected to impact all ecosystems in the

Corresponding author
Emily G. Lim,
emily.lim13@alumni.ubc.ca,
emily.lim13@gmail.com

ocean (*Gattuso et al., 2015*) and past declines in ocean pH have been associated with major extinction events (*Hönisch et al., 2012*), ocean acidification is expected to be one of the most significant factors driving ecosystem change this century (*Caldeira & Wickett, 2003*).

To fully comprehend the ecological implications of OA, it is vital to understand not only the individualistic responses of various species as determined by their physiological and adaptive capacities, but also the responses of communities and ecosystems as species-specific responses intertwine through interspecific interactions and other ecological processes (*Gattuso et al., 2015*). Early OA research focused primarily on direct effects on individual species by measuring metrics such as survival, calcification, growth, development and abundance of individuals (*Kroeker et al., 2013*). Calcified species have been the focus of many studies, as $CO_2$ driven acidification also results in decreased carbonate ($HCO_3^-$) saturation, which negatively affects the ability of species to create calcium carbonate (*Beniash et al., 2010*). In general, heavily calcified taxa like corals, sea urchins, and mollusks are most severely affected by OA, while taxa that rely less on calcium carbonate in their skeletal structures, like crustaceans, are less vulnerable to direct effects (*Guinotte & Fabry, 2008*). OA has also been shown to be metabolically demanding as organisms expend energy compensating for the effects of OA, leaving less energy for other important biological functions whether or not they are calcified (*Wood, Spicer & Widdicombe, 2008*).

These direct effects of OA at the organismal level can be magnified, minimized, or even reversed by indirect effects mediated by interactions with other species (*Kroeker, Micheli & Gambi, 2013*; *Rossoll, Sommer & Winder, 2013*). Because of the potential importance of these indirect effects, many interspecific interactions such as parasitism (*MacLeod, 2017*), herbivory (*Alsterberg et al., 2013*), predation (*Ferrari et al., 2011*), and other interactions (*Rossoll, Sommer & Winder, 2013*) have been incorporated in OA studies. Recently, positive interactions have also been considered in the context of OA. Specifically, changes in biogenic habitat complexity have been suggested as an important pathway through which OA can indirectly affect species (*Connell et al., 2017*), and OA-induced losses of structural complexity in coral reefs, mussel beds, and some macroalgal habitats may result in losses of associated biodiversity (*Fabricius et al., 2014*; *Sunday et al., 2017*). In many cases, the importance of indirect effects may outweigh that of direct effects. For example, although OA has been shown to reduce predator avoidance in fish, *in situ* acidification was shown to increase the abundance of biogenic habitat, leading to an increased abundance of fish despite the impairment of anti-predator behaviours (*Nagelkerken et al., 2016*). Species that have been shown to be physiologically resilient to OA can still face strong indirect effects (*Whiteley, 2011*) that shouldn't be ignored (*Duarte et al., 2016*). Because these ecologically-mediated indirect effects can be just as if not more important than direct effects, they must be considered in tandem with direct effects despite being more difficult to quantify (*Humphreys, 2017*). Neglecting to take indirect effects into account could result in over or underestimating the full effects of OA.

We investigated the potential direct and indirect effects of OA on a marine hydroid—amphipod association in order to fully assess vulnerability to OA. Amphipods, and crustaceans in general (see *Kroeker et al., 2013*), are not expected to be particularly sensitive to the direct effects of OA. Indeed, elevated carbon dioxide is often associated with increased

numbers of amphipods (*Heldt et al., 2016*; *Vizzini et al., 2017*). The increased abundance of amphipods found in these studies may be the result of indirect effects—either higher food supply, reduced competition with other taxa, or reduced predation pressure. Altered biogenic habitat quality or quantity may also play a role, although the effects of OA on the role of biogenic habitat remains poorly studied. Nevertheless, many amphipods are known to rely on biogenic habitat (*Hacker & Steneck, 1990*), which makes them potentially susceptible to the indirect effects of habitat modification (*Sotka, 2007*).

Our amphipod species of interest, the "skeleton shrimp" *Caprella laeviuscula* and *Caprella mutica*, are thought to be strongly associated with biogenic habitat in marine fouling communities (*Caine, 1980*). Fouling communities, comprised of the sessile invertebrates, benthic algae, and associated species that colonize anthropogenic structures, are experimentally tractable systems that have been observed to become less structurally complex under elevated $CO_2$ (*Brown, Therriault & Harley, 2016*; *Brown et al., 2017*). For example, experimental acidification significantly reduced the cover of an important habitat forming species—the hydroid *Obelia dichotoma*—in fouling communities in coastal British Columbia, Canada (*Brown, Therriault & Harley, 2016*). Many species, including caprellid amphipods, have been found living in *O. dichotoma* (*Standing, 1976*) and this reduction of habitat complexity could provide an indirect pathway through which OA would indirectly affect these habitat-dependent species.

In this study, we used caprellid amphipods associated with *O. dichotoma* to explore the direct, physiological effects of OA as well as the potential indirect, habitat-mediated effects of OA. Specifically, we conducted a laboratory experiment to assess the direct effect of OA on caprellid physiology as indicated by changes in heart rate. Further, we observed caprellid habitat preferences in the field, and experimentally manipulated *O. dichotoma* habitat complexity in order to quantify the potential indirect consequences of the predicted OA-driven reduction of *O. dichotoma*. We hypothesized that caprellids would be unaffected by OA directly, as has been observed generally for crustaceans (*Kroeker et al., 2013*), but that they would be negatively indirectly affected through the loss of biogenic habitat complexity. Overall, we predicted that the potential indirect effects of OA could negatively impact caprellid populations even if caprellids were relatively robust in the face of direct effects.

## METHODS

### Field site and study organisms

We collected organisms and conducted field studies at the Reed Point Marina in Port Moody, British Columbia (49°17′31″N, 122°53′25″W) between June 2017 and September 2017. Reed Point Marina is located near the eastern terminus of Burrard Inlet, which experiences seasonal salinity fluctuations in conjunction with spring increases of freshwater input from melting snow, precipitation, and stream input (*Thomson, 1981*). We found two species of caprellid amphipod at this site: *Caprella laeviuscula* Mayer, and *Caprella mutica* Schurin, 1935. *C. laeviuscula* is a common species native to the West coast of North America. *C. mutica* is an introduced species from Japan, and is considered one of the

most widely distributed amphipod species, mostly due to introduction and subsequent spread beyond its native range (*Nakajima & Takeuchi, 2008*). Both species are found in fouling communities composed of colonial hydroids, mussels, tunicates, and bryozoans (*Caine, 1978*; *Caine, 1980*; *Brown, Therriault & Harley, 2016*). Fisheries and Oceans Canada approved organism collections (license number XMCFR 18 2017).

## Effects of ocean acidification on heart rate

To determine the direct physiological effects of ocean acidification on caprellid heart rate, we conducted a 72-h acidification experiment at the University of British Columbia. We collected *C. mutica* and *C. laeviuscula* from Reed Point Marina and transported them to the lab in Tupperware containers filled with seawater. We placed twelve aquaria in a temperature controlled sea table at 17 °C, which was similar to the water temperature at the collection site. We placed a plastic beaker with mesh windows in each aquarium to protect caprellids from mechanical damage from aeration, and added two approximately five-centimeter-long colonies of a hydroid (*O. dichotoma*) for habitat. Seawater pH did not vary between the beaker in which the caprellids were held and the surrounding water in the aquaria, indicating that the mesh windows allowed for adequate water flow between the inside and outside of the beakers. We haphazardly distributed caprellids between the twelve beakers, with 14–16 caprellids in each. Because pilot studies revealed that assessing the heart rate of females was nearly impossible due to the presence of a brood pouch, we used only males in these experiments. Caprellids were acclimatized to these holding conditions for 15 h prior to the initiation of the $CO_2$ manipulation to minimize shock from novel abiotic conditions.

We selected three treatment conditions ($n = 4$ replicate tanks per treatment) that spanned a realistic range of pH values currently observed or predicted for the Strait of Georgia, British Columbia: an ambient control (pH = 8.3), an intermediate pH treatment (pH = 7.9), and a low pH treatment (pH = 7.5). The control treatment pH was representative of average daytime field conditions ($8.25 \pm 0.14$, mean $\pm$ standard deviation), based on weekly pH measurements from the site of organism collection spanning ten weeks from June 28th 2017 to September 7th 2017. The intermediate pH treatment is within the range of predictions of surface ocean pH decline for the end of the century from the IPCC report (*Stocker et al., 2013*). The low pH treatment, while aggressive for open ocean surface water, is well within the range observed in near-surface water in the Strait of Georgia during periods of upwelling (*Riche, Johannessen & Macdonald, 2014*). We established the intermediate and low pH treatments by adding an enriched $CO_2$ mix (40% $CO_2$, 60% air; Praxair) into the aforementioned ambient air lines using mass flow controllers (Smart-Trak; Sierra Instruments, Monterey, CA, USA). Ambient pH conditions were maintained by aerating tanks with ambient air drawn from outside the building.

We took pH (OAKTON pH 150 pH meter calibrated with NBS buffers), temperature and salinity (YSI Pro 30 salinity and temperature meter) measurements daily. In the middle of the 72-hour exposure period we collected water samples and preserved them with mercuric chloride for dissolved inorganic carbon (DIC) analysis (DIC Analyzer model AS-C3; Apollo SciTech Inc., Bogart, GA, USA) following procedures from (*Dickson, Sabine & Christian,*

**Table 1  Seawater chemistry of the OA experiment.** Seawater chemistry of the ambient, intermediate, and low pH treatments ($n = 4$ replicate tanks per treatment) used in the OA lab experiment (mean ± SE).

| Seawater parameter | Ambient pH | Intermediate pH | Low pH |
|---|---|---|---|
| Temperature (°C) | 17.2 ± 0.1 | 17.1 ± 0.1 | 17.1 ± 0.1 |
| Salinity | 27.9 ± 0.02 | 27.9 ± 0.02 | 28.0 ± 0.1 |
| $pH_{NBS}$ | 8.29 ± 0.01 | 7.84 ± 0.01 | 7.54 ± 0.00 |
| Dissolved inorganic carbon ($\mu mol\ kg^{-1}$) | 1,847.73 ± 7.46 | 1,963.34 ± 5.45 | 1,989.63 ± 7.66 |
| $pCO_2$ ($\mu atm$)[a] | 262.00 ± 1.09 | 817.68 ± 8.86 | 1,704.33 ± 6.09 |
| Alkalinity ($\mu mol\ kg^{-1}$)[a] | 2,093.78 ± 8.27 | 2,037.32 ± 6.43 | 1,979.85 ± 8.09 |
| $HCO_3^-$ ($\mu mol\ kg^{-1}$)[a] | 1,661.25 ± 6.61 | 1,862.72 ± 4.98 | 1,892.70 ± 7.42 |
| $\Omega$ Calcite[a] | 4.43 ± 0.02 | 1.78 ± 0.03 | 0.88 ± 0.01 |
| $\Omega$ Argonite[a] | 2.80 ± 0.01 | 1.12 ± 0.02 | 0.56 ± 0.01 |

**Notes.**
[a] Calculated values in CO2SYS (*Pierrot, Lewis & Wallace, 2006*).

*2007*). Measured pH, DIC, temperature and salinity values were used to calculate the remaining seawater carbon chemistry parameters, using the CO2-SYS program (*Pierrot, Lewis & Wallace, 2006*).

The ambient treatment maintained a pH of 8.29 ± 0.01, whereas bubbling air mixed with $CO_2$ successfully decreased pH by 0.45 ± 0.00 units in the intermediate pH treatment, and by 0.74 ± 0.01 units in the low pH treatment. These pH levels correspond to $CO_2$ concentrations of 262.00 ± 1.09 $\mu$ atm, 817.68 ± 8.86 $\mu$ atm, and 1,704.33 ± 7.55 $\mu$ atm (see Table 1 for complete carbonate chemistry). Temperature and salinity remained constant over the course of the 72 h and were similar among treatments (Table 1).

After the 72 h of exposure to treatment conditions, we quantified caprellid heart rate under a dissecting microscope. Heart rate was chosen as a proxy for performance based on previous work quantifying the physiological tolerance of crustaceans to abiotic variables (*DeFur & Mangum, 1979*; *McGaw et al., 2018*). Approximately five to seven individuals of each species were haphazardly selected from each of the twelve aquaria, resulting in a total of 26 *C. laeviuscula* and 21 *C. mutica* in the ambient pH treatment, 19 *C. laeviuscula* and 29 *C. mutica* in the intermediate pH treatment, and 27 *C. laeviuscula* and 21 *C. mutica* in the low pH treatment (48 caprellids per treatment). We transferred the caprellids from the aquaria into small Tupperware containers filled with treatment water, and let them warm to 18 °C for approximately an hour to ensure all caprellids would be filmed at the same temperature. Then, we transferred an individual caprellid on a small piece of *O. dichotoma* from the container into a petri dish filled with treatment water using forceps. Just before placing the petri dish under the dissection microscope we drained the water, leaving only enough to surround the caprellid, which minimized caprellid movement. We found during preliminary work that caprellid heart rate was unchanged after being left in these conditions for about five minutes (a longer period than any of the experimental caprellids experienced), which indicates that if they were experiencing hypoxia it did not affect their heart rate. We mounted an iPhone 5s (version 10.2.1, Apple) to the eyepiece by gluing a cardboard tube to an iPhone case (LifeProof) in order to hold the camera of the phone steady against the eyepiece (Fig. 1A). This allowed us to film the heart beat visible in

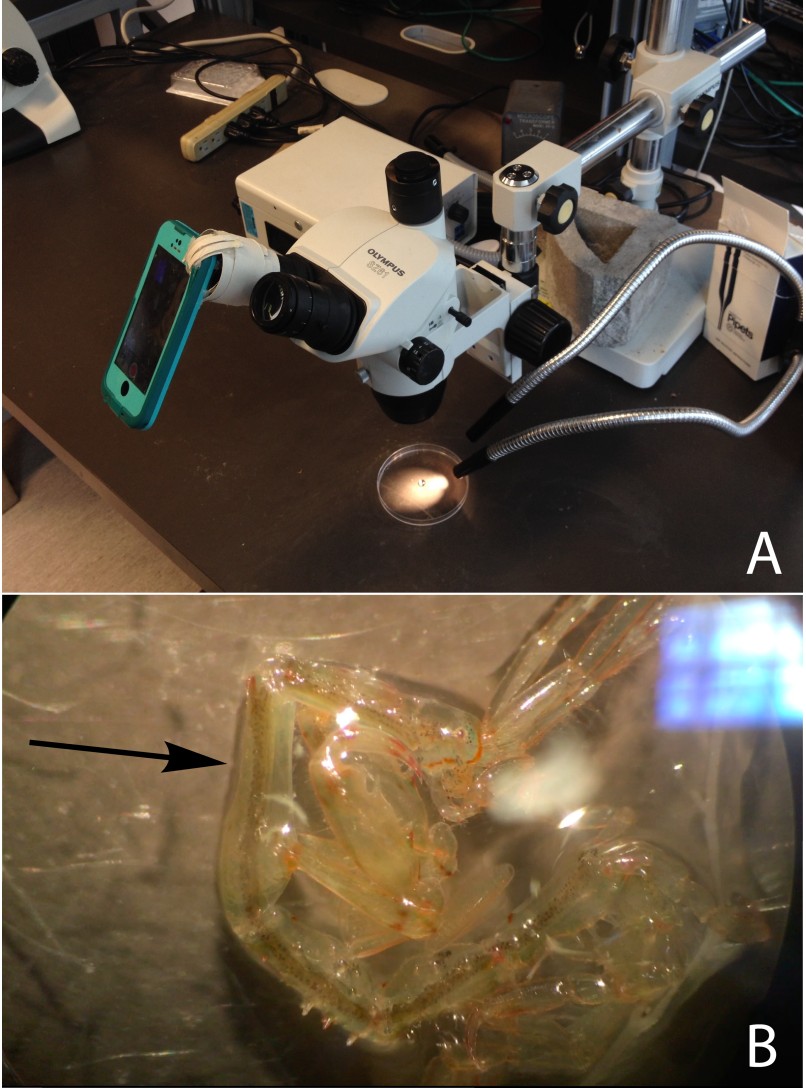

**Figure 1** **(A) The caprellid heart rate monitoring set up and (B) sample footage recorded by the smart phone.** An iPhone was secured to a dissection microscope using a cardboard tube glued to a phone case in order to film the caprellid's heart rate. The black arrow indicates where the heart rate was most clearly visible on the pictured *C. mutica*. Photo credit: (A) Christopher Harley and (B) Emily Lim.

the second pereonite (Fig. 1B), which we did for a minimum of 20 s. Next, we used calipers to measure the distance from the end of the abdomen to the tip of the peduncles of the first antennae, and recorded the species and size. Videos were edited down to approximately 20 s in iMovie (version 10.1.7, Apple) and then played at half speed in order to accurately count heart beats. For each video, we estimated the number of heart beats twice and took the average count in order to ensure accuracy. The two estimates were generally quite close.

## Biogenic habitat preference

We hung twenty 60 cm long, 6 mm thick Nylon ropes weighted with bricks from a dock at Reed Point Marina in July 2017, and allowed fouling communities to develop on them. After ten weeks, we randomly chose ten ropes to assess the number of caprellids on each of the three biogenic habitats abundantly available on the ropes: tunicates (*Botryllus schlosseri*), mussels (*Mytilus trossulus*) and hydroids (*O. dichotoma*). For each habitat type, we assessed a seven-centimeter section of the rope that was fully covered by the habitat type, as this was the largest length of rope that was consistently covered entirely by a single fouling species. *B. schlosseri* is not structurally complex and is difficult to remove without damage, so we counted the number of caprellids (not identified to species) on this species *in situ*. Caprellids on *M. trossulus* and *O. dichotoma* are more difficult to count in the field, and these habitats are easier to remove, so we brought the samples back to the laboratory to count the number of caprellids present under a dissecting microscope.

## Effects of *O. dichotoma* habitat availability

On July 11th, 2017, we collected colonies of *O. dichotoma* from Reed Point Marina. We haphazardly aggregated the *O. dichotoma* colonies into clumps of either one, two, three or four colonies, and then wrapped rubber tubing and a zip tie around the base of each clump to securely attach it to a brick, which we then hung from the dock with rope. All caprellids were removed from the *O. dichotoma* clumps at the beginning of the experiment. The hydroid density treatments were arranged in a randomized block design, with one replicate of each of the four treatments randomly positioned within each of five blocks. The *O. dichotoma* clumps were each one meter apart along the shaded side of a single dock, a location where we observed abundant caprellids. The *O. dichotoma* colonies were left to accumulate caprellids for 8 weeks, after which we brought the clumps of *O. dichotoma* to the lab for analyses. We photographed the individual *O. dichotoma* clumps in a Tupperware container against a white background, and counted and identified the caprellids on each clump under a dissecting microscope. We used Fiji ImageJ to trace the *O. dichotoma* stolons in the images, and then using the ruler photographed for scale we calculated total stolon length.

## Statistical analysis

To analyze the heart rate data from the ocean acidification tolerance experiment, we fitted a mixed effect linear model to the data. Independent variables included treatment, species, size, the species × size interaction, and the random effect of tank nested within treatment. A preliminary model was constructed taking all interaction terms into effect, but as all other interactions were insignificant ($p > 0.70$ in all cases), they were excluded from the final analysis. The heart rate data for *C. laeviuscula* and *C. mutica* were also analyzed separately, by fitting a mixed effect linear model to the data with the treatment and size as fixed effects and tank nested within treatment included as a random effect. A preliminary model showed that the interaction term between treatment and size was not significant ($p > 0.60$), so it was excluded from final analysis.

We log (x+1) transformed the habitat preference data to improve the distribution of residuals. A preliminary analysis indicated that the (random) effect of rope identity was

not significant, so it was omitted. The data transformation failed to meet the assumption of normality, therefore we analyzed the data with a non-parametric Wilcoxon/Kruskal-Wallis Test to compare caprellid abundances among habitats.

In the field manipulation of *O. dichotoma*, the clumps were initially assigned to categorical treatments based on how many colonies of *O. dichotoma* were included in each clump. However, variation in *O. dichotoma* stolon length per colony and variable amounts of growth over the course of the experiment resulted in a quasi-continuous distribution of habitat availability across replicates. Therefore, we treated the total length of *O. dichotoma* stolons per clump as a continuous variable. We log transformed the total number of caprellids to improve linearity, and we performed a regression with a random block term in order to statistically analyze our results. This regression was performed twice, once with just *C. laeviuscula* data (also log transformed), and again with total caprellid data. *C. mutica* was not sufficiently abundant to perform a separate analysis on this species alone.

## RESULTS

### Ocean acidification effects on heart rate

Experimental acidification had a significant effect on caprellid heart rate, with lower pH values resulting in increased heart rate (ANOVA, $F_{2,9.20} = 10.2$, $p = 0.005$; Figs. 2A, 2B). Although heart rate was unrelated to the main effect of caprellid size (ANOVA, $F_{1,136.30} = 0.562$, $p = 0.455$), there was a significant size × species interaction (ANOVA, $F_{1,136.10} = 5.77$, $p = 0.018$) as *C. laeviuscula* heart rate was relatively independent of body size while *C. mutica* heart rate tended to decline with increasing body size (Figs. 2C, 2D). The main effect of species was also significant (ANOVA, $F_{1,135.60} = 25.6$, $p < 0.001$); *C. mutica* had a higher average heart rate (roughly 1.25 times higher) than *C. laeviuscula*, at least in the smaller size classes for both species.

When analyzing the two species separately, we found that a reduction in pH significantly increased heart rate for *C. laeviuscula* from $171.7 \pm 4.45$ beats/min (mean ± standard error) in the ambient treatment to $182.2 \pm 5.19$ beats/min in the intermediate pH treatment and $195.1 \pm 5.56$ beats/min in the low pH treatment (ANOVA, $F_{2,8.25} = 7.19$, $p = 0.016$), but found no effect of caprellid size on heart rate (ANOVA, $F_{1,67.34} = 1.365$, $p = 0.247$). Looking only at *C. mutica*, we found that decreased pH significantly increased heart rate from $202.1 \pm 8.36$ beat/min in the ambient treatment to $203.1 \pm 5.41$ beats/min in the intermediate pH treatment and $228.0 \pm 7.84$ beats/min in the low pH treatment (ANOVA, $F_{2,5.21} = 5.99$, $p = 0.045$) and found a negative linear relationship between caprellid size and heart rate (ANOVA, $F_{1,63.28} = 15.7$, $p < 0.001$).

### Biogenic habitat preference

We found that caprellid abundance differed significantly between habitat types in the field (Wilcoxon Test, $DF = 2$, Chi Square = 22.2, $p < 0.001$; Fig. 3). Caprellids were most abundant on the hydroid *O. dichotoma*, while mussels (*Mytilus trossulus*) and tunicates (*Botryllus schlosseri*) hosted less than 5% of the individuals found in *O. dichotoma*.

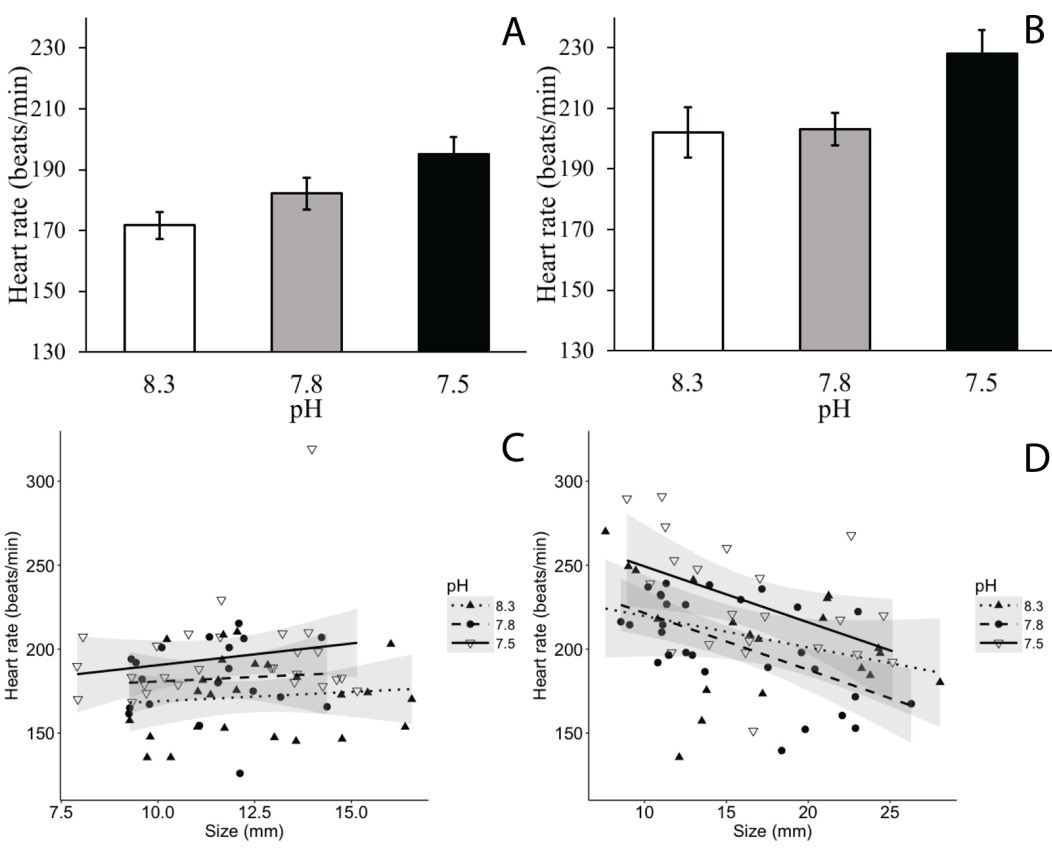

**Figure 2  Response of caprellid heart rate to ocean acidification.** Response of (A) *C. laeviuscula* and (B) *C. mutica* heart rate to ambient (pH = 8.3) intermediate (pH = 7.8) and low (pH = 7.5) pH levels, pooled across body sizes. Error bars represent standard error ($n = 4$ replicate tanks per treatment). Effect of size and ambient (pH = 8.3), intermediate (pH = 7.8), and low (pH = 7.5) pH treatments on (C) *C. laeviuscula* and (D) *C. mutica* heart rate. Each point represents the heart rate of a single caprellid, and the shaded area represents 95% confidence intervals. Decreased pH significantly increased heart rate for *C. laeviuscula* ($p = 0.016$) and *C. mutica* ($p = 0.045$). Body size had a significant effect on heart rate for *C. mutica* ($p < 0.001$), but not for *C. laeviuscula* ($p = 0.247$).

### Effect of *O. dichotoma* density

We found a significant positive relationship between the total *O. dichotoma* stolon length and the number of *Caprella* spp. per habitat clump (linear regression: $R^2 = 0.769$, $F_{1,13.08} = 15.9$, $p = 0.002$; Fig. 4). The vast majority of these caprellids were *C. laeviuscula*; when the abundance of this species was analyzed alone, a similar positive relationship with habitat availability was found (linear regression, $R^2 = 0.731$, $F_{1,13.35} = 13.4$, $p = 0.003$; data not shown). As we only found five *C. mutica* individuals, all of which were male, we were unable to statistically analyze patterns for this species.

## DISCUSSION

Ocean acidification, like all environmental change, can affect ecological processes in a number of ways. The proximate effects are direct, mediated by the physiology of impacted

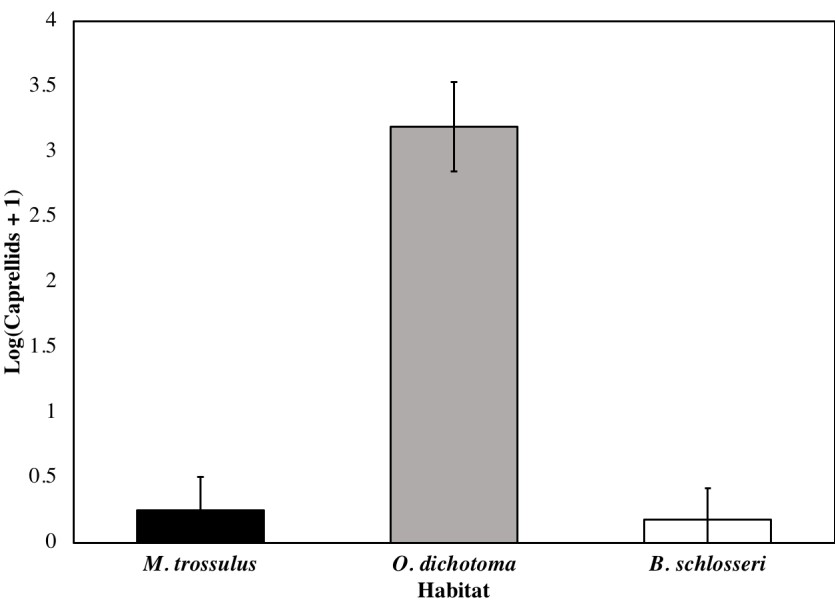

**Figure 3  Average abundance of caprellids on mussels (*Mytilus trossulus*), hydroids (*O. dichotoma*), and tunicates (*Botryllus schlosseri*).** The number of caprellids significantly differed between habitat types ($p < 0.001$). Error bars represent standard error ($n = 10$).

species and their capacity to tolerate, acclimatize, or adapt to changing conditions. These physiological changes can result in differences in performance, which can affect per capita interactions (e.g., feeding rates), and differences in growth, reproduction, and survival, which, over the long term, can result in changes in population size. Because species in a community are joined in a complex web of interactions, any environmentally-driven changes in per capita interaction strength or population size can trigger substantial ecological changes via indirect pathways. Therefore, it is critical to incorporate both direct and indirect effects in studies of ecological responses to environmental change.

In our system, changes to a major habitat forming species (*O. dichotoma*) have been measured in response to OA (*Brown, Therriault & Harley, 2016*), so this seemed like a likely avenue for indirect effects on caprellid amphipods if they are indeed reliant on *O. dichotoma* as habitat. We confirmed that vastly more caprellids were found on *O. dichotoma* than on *Mytilus trossulus* and *Botryllus schlosseri*, which substantiates a habitat preference found in previous studies (*Caine, 1980*). The relatively small number of caprellids that were found on *M. trossulus* and *B. schlosseri* appeared to be in transit, as opposed to the caprellids found on *O. dichotoma*, which were more often secured in place. This suggests that the caprellids on *M. trossulus* and *B. schlosseri* may have been crossing these species to reach a different habitat type, as opposed to living on them, suggesting that the differences we observed may actually be conservative. There are three possible reasons that caprellids were more abundant on *O. dichotoma*. First, C. *laeviuscula* is dependent on periphyton/detritus as its major food supply (*Caine, 1980*), and C. *mutica* has been observed scraping feeding as well (*Cook, Willis & Lozano-fernandez, 2007*). Tunicates do not accumulate large amounts of

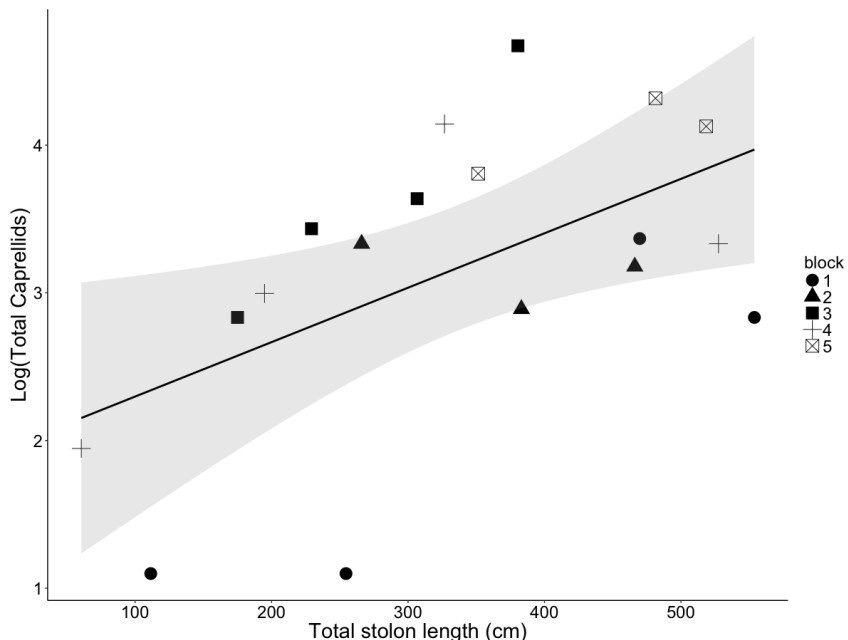

**Figure 4** **Response of caprellid population size to increased total stolon length of *O. dichotoma* per clump.** The line of fit indicates the linear relationship between caprellid population size and *O. dichotoma* habitat availability, represented by the total *O. dichotoma* stolon length of each habitat clump ($p = 0.002$). Each point indicates the size of the caprellid population on a single clump of *O. dichotoma*, and the symbol indicates which block the clump was randomly placed in. The single regression line ignores the random block effect for the sake of simplicity, although the effects of both total stolon length and block ($n = 5$) were significant. The shaded area indicates the 95% confidence interval.

periphyton, and other grazers such as gastropods keep mussels clean (*Fricke et al., 2016*). It is possible that more periphyton accumulates on the stolons of hydroids, making them a better patch for foraging. Second, the thin stolons of *O. dichotoma* provide excellent attachment points for the pereopods of caprellids, and less complex habitat may be more difficult to anchor to. Finally, caprellids visually resemble *O. dichotoma*, and this camouflage may be an important determinant of their field distribution via direct habitat selection by the caprellids, disproportional rates of predation on caprellids that occupy mussels or tunicates, or both.

Under OA, *O. dichotoma* abundance is projected to decrease, providing a pathway by which we expected OA to indirectly affect caprellids via shifts in biogenic habitat (*Brown, Therriault & Harley, 2016*). Although elevated $CO_2$ has been shown to cause an increase in some amphipod populations (*Heldt et al., 2016*; *Vizzini et al., 2017*), the amphipods in these studies were associated with biogenic habitats proposed to benefit from OA, as opposed to *O. dichotoma*, which is not. When we experimentally manipulated the amount of available *O. dichotoma* habitat, we found a positive relationship between *O. dichotoma* abundance and caprellid population size, supporting our hypothesis that caprellid population size is dependent on *O. dichotoma* abundance. These results, in conjunction with those of *Brown, Therriault & Harley (2016)*, suggest that under OA, indirect effects of habitat modification

will act to significantly reduce caprellid populations. Similarly to previous studies, our results indicate that the abundance of amphipods is strongly tied to the success of their biogenic habitat in the context of OA. In cases where OA has a positive effect on biogenic habitat such as fleshy macroalgae, a positive effect is seen on amphipod populations (*Kroeker et al., 2011*). In our case, OA is projected to negatively affect *O. dichotoma* and as such we predict that caprellid populations will decline as well.

There are several mechanisms that could drive the dependence of caprellid population size on *O. dichotoma* availability. Primarily a scraper feeder, *C. laeviuscula* is quite dependent on periphyton accumulation, although they have been observed filter feeding as well (*Caine, 1980*). Studies have also documented *C. mutica* both filter feeding and scraping, but the exact feeding preference remains unknown (*Cook, Willis & Lozano-fernandez, 2007*; *Nakajima & Takeuchi, 2008*). If periphyton accumulation is restricted by the amount of available substrate, it is possible that smaller populations of caprellids were found on smaller clumps of *O. dichotoma* due to food limitation. Another possible mechanism that could explain the relationship between caprellid population size and *O. dichotoma* abundance is interference competition for space. *C. laeviuscula* and *C. mutica* have both been shown to be aggressive to other caprellids (*Caine, 1980*; *Shucksmith et al., 2009*), and it is possible that less extensive *O. dichotoma* clumps were less populated due to overcrowding and resultant interference competition. The final mechanism that could be responsible for the linear relationship between caprellid populations and *O. dichotoma* is predation. Complex *O. dichotoma* could be acting as a refuge for caprellids to hide from visual predators such as fish, so less refuge would result in an increased mortality due to predation. We observed much larger caprellid populations in a fouling experiment conducted in fish-free flow through mesocosms over the previous summer, which suggests that predator exclusion may have an effect on caprellid population size (E. Lim & N. Brown, 2017, unpublished data). The true mechanism may be a combination of some or all of the aforementioned mechanisms, all contributing to the observed dependence of caprellid population size on *O. dichotoma*.

Owing to the strong association between caprellids and *O. dichotoma*, our research suggests that the reduction of *O. dichotoma* due to OA will result in a proportional decline in caprellid populations, which might point to similar trends in other species that rely on *O. dichotoma* for habitat. Colonial hydroids such as *O. dichotoma* have been found to support a large variety of species, including other amphipods, gastropods, and polychaete worms (*Round et al., 1961*). These are all organisms that could potentially face population declines as *O. dichotoma* abundance decreases under OA. Furthermore, caprellid amphipods make up an important component of the diet of fishes (*Woods, 2009*), so caprellid population declines could have consequences for their fish predators as well.

While we expected to find indirect effects of OA on caprellids, we did not expect to find direct effects. In general, crustaceans are physiologically resilient to OA when considering the effects of OA on survival, calcification, growth, development, and abundance (*Kroeker et al., 2013*). This is due to the composition of their exoskeletons, and their ability to compensate for internal acid–base disruptions (*Whiteley, 2011*). Studies looking specifically at the response of embryonic crustacean heart rate to OA have found no effects (*Styf, Sköld*

*& Eriksson, 2013*; *Schiffer et al., 2014*). Therefore, although the observed relationship between body size and heart rate in *C. mutica* was unsurprising given past observations and the tenets of metabolic theory (*DeFur & Mangum, 1979*; *Brown et al., 2004*), the significant increase in heart rate under acidified conditions for both species was unexpected. Heart rate is a general proxy for metabolic rate, and may thus indicate higher basal metabolic costs in acidified conditions (*Wood, Spicer & Widdicombe, 2008*), though whether we observed a stress or metabolic effect is unclear. While it is rare to find a direct effect of OA on crustaceans in general, OA has been shown to decrease the survival of some amphipods (*Poore et al., 2013*; *Schram et al., 2016*), and some metabolic effects have been found with respect to juvenile isopods (*Turner et al., 2016*). Conversely, OA had no effect on the growth rate of another amphipod species *Gammarus locusta* (*Hauton, Tyrrell & Williams, 2009*). These results indicate that further consideration of the effects of OA on amphipods is necessary before we are able to generalize broadly across this grouping.

Although it remains unclear if the physiological effects of OA would translate into changes in individual performance or population dynamics for *Caprella* spp., it is worth considering this possibility. If the increase in heart rate does in fact correspond to a decreased population size or shifted individual performance, this might produce an ecological feedback to their *O. dichotoma* habitat. Caprellids have a strongly negative effect on epiphytes, and in the absence of caprellids, periphyton biomass increased by 411% in *Zostera marina* beds (*Caine, 1980*). If individual caprellid performance is compromised, there could be increases in diatom fouling on *O. dichotoma*, which could affect both *O. dichotoma* and food availability in the community. Although we only explicitly considered the importance of indirect effects on caprellids as mediated by another species (*O. dichotoma*), it is equally possible that changes in *Caprella* spp. abundance or feeding rates could mediate indirect effects of OA on other species.

It remains to be seen if our finding of negative direct effects on amphipods is general, but it is certain that our overall prediction of decreased population size for caprellids in our study system will not apply to all species in all situations given context dependence in the combinations of direct and indirect effects at work. For example, findings that amphipod abundance were elevated at low pH sites in the Mediterranean (*Scipione, 2013*; *Scipione et al., 2017*) could be the result of positive indirect effects swamping any negative direct effects. Elevated $CO_2$ has also been linked to increased caprellid populations near $CO_2$ vents, and it has been suggested that competitive release is the underlying mechanism (*Cigliano et al., 2010*; *Ricevuto et al., 2012*). In these cases, the positive indirect effects of OA are able to overwhelm any direct negative physiological effects. Conversely, the negative direct and indirect effects of OA that we found may act in concert. If caprellid metabolism increases under OA, and their refuge from predators and source of accumulating periphyton is reduced in overall abundance, perhaps the food limitation could negatively affect caprellids even more once they also have an increased resting metabolic rate. These results stress the importance of considering indirect effects on amphipod species and other small grazers, particularly the tight relationship between such species and their habitat, in combination with any direct physiological consequences of environmental variation.

## CONCLUSION

Both direct and indirect effects are important in determining the ultimate ecological responses to patterns of environmental change. We found that caprellid amphipods are directly physiologically affected by OA, and are also likely to be indirectly affected through an OA driven decline in habitat complexity. Therefore, we predict that caprellid populations will decline as a result of ocean acidification in our study area. Other *Obelia*-dependent species may suffer similar population declines, which would in turn have implications for biodiversity of fouling communities as a whole. As may be the case in other systems dominated by biogenic habitats (*Sunday et al., 2017*), shifts in biogenic habitat complexity may be the main driver of caprellid population decline under OA. However, this habitat-dependence alone could under-predict the full effects of OA if there is also a direct metabolic cost for habitat-dependent species.

## ACKNOWLEDGEMENTS

We are grateful to everyone in the Harley Lab for all of their support and encouragement. We thank Norah Brown for sparking an interest in caprellids, and Anna Doebeli for her assistance in the field. We thank Amelia Hesketh, Kathryn Anderson, and Colin MacLeod for their help with the $CO_2$ system and calculations related to seawater chemistry, and Cassandra Konecny for her overall assistance. Rod MacVicar and the Reed Point Marina staff generously allowed us the use of their docks. Special thanks are owed to EGL's mother Linda Greenway, for suffering through first draft revisions, "borrowed" Tupperware containers and coolers, and caprellids in her fridge. Finally, we would like to thank Iain McGaw and an anonymous reviewer for their comments and suggestions.

### Funding

This research was supported by a University of British Columbia Science Undergraduate Research Experience Award to Emily G. Lim and by a Natural Sciences and Engineering Research Council Discovery Grant to Christopher D.G. Harley. The funders had no role in study design, data collection and analysis, decision to publish, or preparation of the manuscript.

### Grant Disclosures

The following grant information was disclosed by the authors:
University of British Columbia Science Undergraduate Research Experience.
Natural Sciences and Engineering Research Council Discovery Grant.

### Competing Interests

The authors declare there are no competing interests.

## Author Contributions

- Emily G. Lim and Christopher D.G. Harley conceived and designed the experiments, performed the experiments, analyzed the data, contributed reagents/materials/analysis tools, prepared figures and/or tables, authored or reviewed drafts of the paper, approved the final draft.

## Field Study Permissions

The following information was supplied relating to field study approvals (i.e., approving body and any reference numbers):

Organism collections were approved by Fisheries and Oceans Canada (license number XMCFR 18 2017).

## Data Availability

The raw data are provided in the Supplemental Files.

## Supplemental Information

Supplemental information for this article can be found online at http://dx.doi.org/10.7717/peerj.5327#supplemental-information.

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
