# Peer review of "Caprellid amphipods (Caprella spp.) are vulnerable to both physiological and habitat-mediated effects of ocean acidification"

_PeerJ, doi:10.7717/peerj.5327_

## Round 0.1 · original submission · Minor Revisions

· Academic Editor

Minor Revisions

Both reviewers liked your article and made a number of constructive suggestions. I look forward to receiving a revised MS that takes the reviews into account.

·

Basic reporting

This paper is well written and a nice little preliminary study on interactive biotic and physiological effects. One could argue that it might need a little more analysis on the heart-rate side of things. That being said it is of note that the authors have worked on a lesser known group of crustaceans for which very little work on heart rate has been performed.

Overall I prefer papers when they are not written in the first person.

Minor points
L70 ..and more (Rossoll,.. more what?
L97 for sentence ending quality

L116 onwards - These are really predictions rather than hypotheses. I would tend to omit these and give a shorter (1-2 sentence aims). Something like the aim was to determine the potential effects of reduced pH on metabolic rate and if and how this may interact with habitat complexity to determine the future distribution of caprellid amphipods

L131 Field studies were conducted, and organisms were collected...

L189 - McGaw et al is 2018, also check - one of the authors last names is incorrect!

Results
Very brief, can you give a bit more detail and guide the reader through the results. For example, in biogenic habitat preference how many more caprellids were found on Obelia - give some details rather than general trends

Figures
Can you indicate significance on the graphs. Also could you include more description in all the legends. I do not understand the C and D figure 1- can you explain what these are and what they mean please. Same for figure 3 can you explain what this is, what the line represents and the shaded area

Experimental design

It is interesting that the authors appear to have used a novel method of a cell phone to record heart rate. But it is only briefly described, this is novel and while would not be great for most applications, it might be very useful for undergraduate projects given the low cost and ease of use. Some more detail would be beneficial and even some screen shots included in the paper as figures.

I do have some concern as to how accurate/useful the rates will be. I would assume the animals would be stressed only being in a small amount of water, but more importantly it would likely become hypoxic, and crustaceans show a strong bradycardia response under such circumstances. The stress response could be reduced, but not abolished, by allowing them to settle in the petri dish for several hours and flow through water would reduce oxygen depletion. The common way to hold these animals is to put a spot of superglue on the carapace and then stick a Q-tip or similar structure to the carapace - keeps the animal in place and they can be manipulated for seeing the heart. I know this may not be feasible at this stage, but keep this in mind for future studies. In addition HR of crustaceans tends to be quite variable between individuals, but also within an individual over even short time periods. Therefore having an N=4, and a short trace of 20sec may not be that reliable if one wants to draw definite conclusions - but it is a nice start and this phone method should be explored further.

The authors state that HR in crustaceans can be used as a proxy for metabolism, but they also will have access to a whole other set of data - stroke volume! By modelling changes in shape of the heart one can calculate the stroke volume of the heart, and thus cardiac output. This is important because HR and SV vary independently of one another to determine how much blood is pumped from the heart. See papers by Harper and Reiber on grass shrimp for methods. It can be measured from a short video clip.

The authors state why they used 17C, but why were animals acclimated for 15h after capture and then 72h to the test pH are these times particularly relevant to anything?

Validity of the findings

This paper makes a nice connection between the biotic (environment) and physiological changes. The heart rate looks like it does change, the authors shouldn't be too surprised by this, whether it is a stress or a metabolic effect/cost is unclear. The authors have used 2 embryonic papers that found no difference in other crustaceans - but this is not to say that there may be an increase in this adult species. There are a lot of physiologically different mechanisms between juveniles and adults. I would try and improve the discussion by strengthening the links, for example if cover disappears maybe the animals would be more stressed or food deprived, how would this affect them given they now also have a higher metabolic rate? At present the two effects are discussed somewhat separately.
It would also be interesting to see what effect long term pH had on growth and feeding ability to draw definite conclusions. Maybe this is something for another study.

Discussion - Ok I would try and tie in the ecological an physiological changes a little more. For example it does appear that metabolic rate is higher in low pH. If then they cover and possible food source is lower what might this do to an animal that is already stressed by low pH.

Overall I enjoyed reading this paper - I liked the way the authors used both physiological and habitat changes to determine possible effects of OA.

Additional comments

A couple of questions that I always pose to this type of predicted environmental data and climate change scare mongering
1. You subjected the animals to a sharp decrease in pH, when in reality pH is going to change gradually over the next 80 years. I would assume that these organisms are reaching maturity quite quickly and there could be tens of generations (if not more) in that time. Given both the hydroids and caprellids may show developmental plasticity, or even short-term evolution, how realistic do you think these experiments are in general? Do you think that most organisms would adapt over several generations to any aspect of climate change?
2. If these organisms do decrease, or even disappear altogether why should we care? Are they important in the big picture of things, what feeds upon them, are they key players in the ecosystem?
IF all these climate changes come to fruition, (and given the previous media hype during the last 50 years about nuclear war, the ozone hole, various flu pandemics, acid rain, meteorites, the Aztec calendar, over population and Uncle Tom Cobbly), I doubt whether they will have any noticeable effect. However, should they come true the actual global social disruption that will occur will mean than no one gives two hoots about some little animals, or even any other animals. They will all be buying guns and running for the hills!
So I am with President Donald Trump on the whole climate change thing!

Reviewer 2 ·

Basic reporting

The paper by Lim and Harley, is a relatively simple and somehow indirect experiment to estimate the synergistic effects of physiological and habitat-mediated aspects of Ocean Acidification (OA) on amphipods caprellids....
This is a quite challanging and interesting topic and the paper, although a bit simplistic, deserve publication. However, before final acceptance it is necessary to consider and expand in the discussion and in the comparisons the following points:
a) discuss and add literature on physiology (respiration and metabolic rate) with particular focus on crustaceans and OA (e.g. Turner et al., 2016 Marine Biology 163, 211)
b) expand and discuss more paper related to abundance pattern of amphipods to OA and their relationships with the habitat (e.g., the macroalgae) (e.g. see es. Scipione, M.B. 2013. On the presence of the Mediterranean endemic Microdeutopus sporadhi Myers, 1969 (Crustacea: Amphipoda: Aoridae) in the Gulf of Naples (Italy) with a review on its distribution and ecology. Mediterranean Marine Science 4, 56–63.
Kroeker, K.J., Micheli, F., Gambi, M.C. & Martz, T.R. 2011. Divergent ecosystem responses within a benthic marine community to ocean acidification. Proceedings of the National Academy of Sciences USA108, 14515–14520.
SCIPIONE M.B., KROEKER K.J, RICEVUTO E., GAMBI M.C. 2017. Amphipod assemblages along shallow water natural pH gradients: data from artificial substrata (Island of Ischia, Italy). Biodoversity Journal, 8(2): 469-470.
Ricevuto, E., Lorenti, M., Patti, F.P., Scipione, M.B. & Gambi, M.C. 2012. Temporal trends of benthic invertebrate settlement along a gradient of ocean acidification at natural CO2 vents (Tyrrhenian Sea). Biologia Marina Mediterranea19, 49–52.)
Poore et al., 2013 Oecologia https://www.ncbi.nlm.nih.gov/pubmed/23673470)

I found in fact that respect to the interest and complexity of the topic more literature could be considered.
this is the only point I warmly suggest, for the rest the paper is well written.

Experimental design

Quite simplistic but correct

Validity of the findings

The results should be better discussed and compared at the light of various papers whihc have not considered and whihc rae related to bot capreilli amphipods and other crustaces, related with OA

---

## Round 0.2 · Minor Revisions

· Academic Editor

Minor Revisions

Thanks for your responses to the reviewers' comments. I'm happy with all of them, and will accept the MS after correction of the following typos etc:

Lines 62, 85, 99, 194, 226, 305, 318, 337: add a comma before the "which"
192: delete "in"
239: fitted
399: epiphytes
426: affected

---

## Round 0.3 · accepted · Accept

· Academic Editor

Accept

Thanks for making the final changes.

#